# Failure Analysis of SAC305 Ball Grid Array Solder Joint at Extremely Cryogenic Temperature

**Yanruoyue Li [1]** , **Guicui Fu [2]**, **Bo Wan [2],\***, **Maogong Jiang [2]**, **Weifang Zhang [2]** and **Xiaojun Yan [1]**

1   School of Energy and Power Engineering, Beihang University, Beijing 100191, China;
    lyry2011@buaa.edu.cn (Y.L.); yanxiaojun@buaa.edu.cn (X.Y.)
2   School of Reliability and Systems Engineering, Beihang University, Beijing 100191, China;
    fuguicui@buaa.edu.cn (G.F.); maogong@buaa.edu.cn (M.J.); 08590@buaa.edu.cn (W.Z.)
*   Correspondence: wanbo@buaa.edu.cn; Tel.: +86-156-5292-8449



**Featured Application: It is an important trend that Pb-free materials in electronic devices/components are used in the aerospace field. Several reliability issues associated with this kind of material characteristic occurred and need in-depth studies. One of the issues is that Sn-rich material's characteristic changes under low-temperature. This change may cause components/device failure and has an influence on their reliability. This work of failure analysis provides an actual case of electronic components with failure when under harsh environments, such as extreme cryogenic temperature in space. It makes reliable use of Pb-free material under extremely cryogenic temperature conditions to be taken seriously, brings up an analysis process for this kind of failure, and suggestions for operating temperatures are put forward.**

**Abstract:** To verify the reliability of a typical Pb-free circuit board applied for space exploration, five circuits were put into low temperature and shock test. However, after the test, memories on all five circuits were out of function. To investigate the cause of the failure, a series of methods for failure analysis was carried out, including X-ray detection, cross-section analysis, Scanning Electron Microscope (SEM) analysis, and contrast test. Through failure analysis, the failure was located in the Pb-free (Sn-3.0Ag-0.5Cu) solder joint, and we confirmed that the failure occurred because of the low temperature and change of fracture characteristic of Sn-3.0Ag-0.5Cu (SAC305). A verification test was conducted to verify the failure mechanism. Through analyzing data and fracture surface morphology, the cause of failure was ascertained. At low temperature, the fracture characteristic of SAC305 changed from ductileness to brittleness. The crack occurred at solder joints because of stress loaded by shock test. When the crack reached a specific length, the failure occurred. The temperature of the material's characteristic change was −70–−80 °C. It could be a reference for Pb-free circuit board use in a space environment.

**Keywords:** SAC305; BGA; low temperature; fracture failure

## 1. Introduction

Because of the importance of space exploration, more and more institutions are turning their research direction to the deep space. Considering the special environmental condition, there may be a variety of composite reliability issues, especially the reliability problems of electronic components operating under low-temperature conditions [1]. It is hard to maintain or replace the equipment operating in space. Once there is a problem, the consequences are difficult to predict. Therefore, it is significant to study the possible failure mechanism of electronic components in the ultra-low temperature environment [2,3].

Failure of the solder joint plays an important role in the field of electronic components' reliability [4]. For a long time, 63Sn-37Pb has become the most appropriate solder material owing to its practicability, economy, and superior performance. Research about Sn-Pb solder has been maturing, and it has been used as the main material in the packaging structure of various electronic components [4,5]. But Pb is a toxic metal and will contaminate the environment. In order to decrease the impact on the environment, the EU issued the Restriction of the Use of Certain Hazardous Substances in Electrical and Electronic Equipment (RoHS) on 13 February 2003, which accelerated the Pb-free process. However, there are some exempt applications put forward in this document [6]. These exceptions include space exploration. But with the rapid development of electronic industry and aerospace technology, traditional Sn-Pb material cannot meet the requirement of high performance and high reliability [5]. Pb-free in the space exploration field is the tendency in the future. Thus, Pb-free solder material has been developed and studied. Component manufacturers are forced to use Pb-free solder instead of 63Sn-37Pb solder [7]. Recently, Sn-Ag-Cu solder has become one of the most useful materials as a replacement of Sn-Pb solder. More studies of this kind of material have been carried out. A considerable amount of practice indicates that there are some differences between Sn-Ag-Cu (SAC) solder and Sn-Pb solder in terms of reliability [8]. With the difference in the amount of Ag and Cu, there are many kinds of Pb-free solder material, such as Sn-3.0Ag-0.5Cu (SAC305), Sn-3.8Ag-0.7Cu (SAC387), Sn-3.5Ag-0.7Cu (SAC357), and SAC with other composition. Researchers not only have made a study on the properties of basic SAC materials but also on the influence of other solder composition on material properties, such as performance of different composition and comparison of different additional quantity [9–11]. Among them, SAC305 is one of the most common Pb-free solder materials. In our actual practice, the very material of our failed Pb-free circuit solder is SAC305 as well.

Aiming at the reliability of Pb-free solder joints, there is a lot of researches and discussions. The main source of the Pb-free solder joint reliability problem is as follows: Shear fatigue and creep crack of solder joints [7,12], electro-migration [5,12], cracks formed by intermetallic compound (IMC) between solder and matrix interface [13,14], the short circuit caused by Sn whisker growth [13], and electric and chemical corrosion [15]. Based on these reliability problems, researchers, such as S. Pin [16], G Jian [17], A Surendar [18], Zijie Cai [19], and F Liu [20] have made related test studies to find out the mechanism of failure. Most of the researches are about temperature cycling, mechanical shock, and electro-migration. Xu Long [15], Liu, XG [21], and M Aamir [22] researched the influence of different material elements added in Pb-free material. In addition, X Niu [23] and D. S. Liu [24] made studies on the low temperature's effect on solder joint fracture behavior. Yet, the lowest temperature in these studies is −45 °C. Further studies are needed to explain the failure occurred under extremely cryogenic temperatures in a space environment.

The aim of this work was to find out the root cause of a failure occurred in practice. The failed component was the memory of a typical Pb-free circuit board, which is used for space application. First, failure analysis methods were used to confirm the failure mechanism. Then, in order to verify the results of failure analysis, samples were designed, as well as put into the verification test. By analyzing the results of low-temperature tensile test for SAC305 solder material, the root cause of this failure case was put forward. Finally, the conclusion was drawn at the end of the manuscript.

## 2. The Subject of Study and Failure Background

Because of the large number of applications of Pb-free components, the Pb-free circuit board used for deep-space applications attracts a lot of interest. At present, users want to know whether the Pb-free circuit board can be used reliably in space.

In order to verify the reliability of a typical Pb-free circuit board in aerospace applications, five circuits were put into low-temperature and shock test, which was called a qualification test in the subsequent section. Combining with application requirements and JEDEC standards (JESD22-B110 and JESD22-A119), the test temperature was −100 °C, and the test acceleration was 100G, 0.5-millisecond duration, and half-sin pulse. After the tests, all five circuits had failed, depending on the results of the

printed circuit board (PCB) function test. There was no output of the circuit. Function tests for every component were conducted, and it could be confirmed that the failure of the PCB was caused by the memory (Figure 1) on the circuit. The memory could not store and read data normally. The package of the memory was Ball Grid Array (BGA), and the solder material was SAC305. Further analysis is needed in order to find out the root cause of the failure.

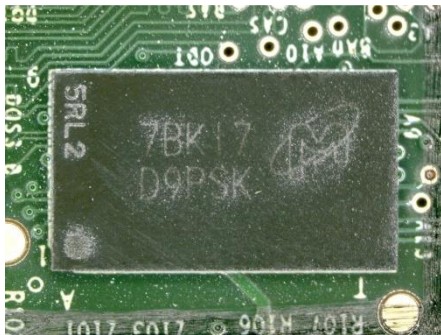

**Figure 1.** The failed memory on the board.

## 3. Failure Analysis of the Memory

A series of methods were used to make the failure analysis. Figure 2 shows the process of failure analysis. The appearance of all five memories was observed under a stereoscopic microscope, and there was no obvious damage on the surface. Two of the five memories were removed to put into the electronic function test. The results showed that the components themselves were intact. Thus, we guessed that the failure occurred at solder joints. When the failure was located at the BGA solder joint, X-ray detection was carried out at first. Second, the cross-section was analyzed using a metallurgical microscope. Then, the solder ball tensile test was conducted, and the fracture surface of the ball was observed using SEM. Finally, a contrast test under room temperature was conducted to find out the failure mechanism.

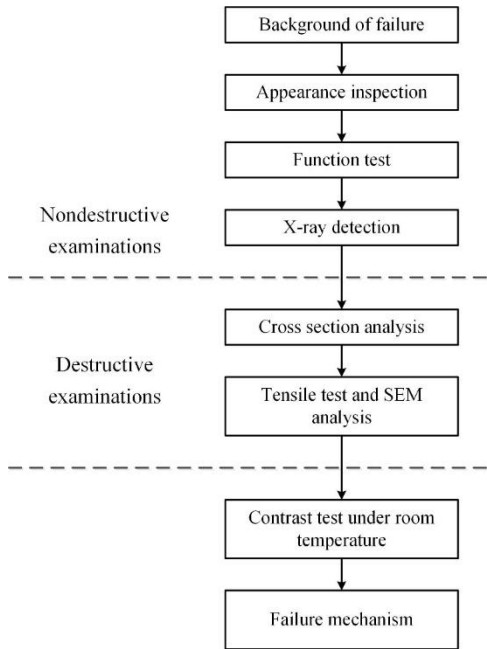

**Figure 2.** The process of failure analysis.

### 3.1. X-ray Detection

X-ray detection is one of the most important nondestructive detection methods used in failure analysis. It can find out defects of solder balls, such as voids, and size inconsistency. Figure 3 shows the BGA of the memory through X-ray. The deformation of solder balls could be seen obviously (the red circle in Figure 3). However, observation of deformation could not explain the failure cause and mechanism. Therefore, further analysis of the deformed solder balls is necessary.

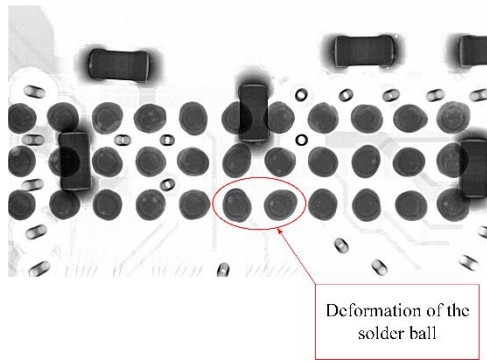

**Figure 3.** X-ray detection shows the deformation of the solder ball.

### 3.2. Cross-Section Analysis

Afterward, a cross-section of solder joints was prepared, and cracks were found under a metallurgical microscope, as shown in Figure 4. It could be seen that the solder ball on the left had a crack almost penetrating it, and the one on the right was broken completely. Combined with the previous conclusion, the memory itself was intact; it indicated that the failure was caused by cracks of the BGA solder joint.

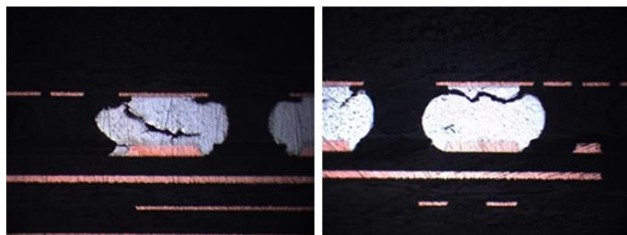

**Figure 4.** Cracks in the solder joint observed under a metallurgical microscope.

### 3.3. Tensile Test and SEM Analysis

Considering the special Pb-free material (SAC305) and the extreme environmental condition, the authors speculated that the failure was relevant to low temperature based on existing researches [25,26]. The other three memories of the failed circuit board were put into the tensile test. The universal tensile testing machine was used to pull the memory off the board. The corresponding fixture was made, and the tensile test was conducted under room temperature with a constant speed of 0.1 mm/min. The fracture occurred on the solder ball. Then, SEM was used to observe the fracture surface of the solder joints. The surface morphology under SEM is shown in Figure 5.

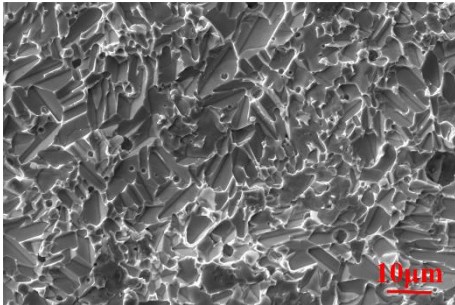

**Figure 5.** The surface morphology of BGA under SEM shows the brittle fracture.

From Figure 5, the obvious intergranular fracture could be seen. It is the characteristic of brittle fracture. It could be observed that there was a brittle fracture of the solder joint. Theoretically, the fracture behavior of SAC305 should be ductile at a normal temperature. Actually, the result of SEM showed that there was a change in material characteristics. According to relative studies, Sn-based solder transforms from ductile to brittle at low temperatures. This is because of an isomer transition phenomenon (commonly known as Sn-pest) of Sn-rich solder alloys when the temperature is low, and for SAC305, the temperature is lower than –30 °C [27]. The fracture changes from β-Sn to α-Sn and the volume of this kind of Sn fracture can expand by 26%, resulting in partial or total fracture of the solder joint [28]. We think the reason is that low temperature causes the change in material characteristics, and it is much easier to fracture under cryogenic and shock tests because of the brittleness. In order to verify our conjecture, a contrast test under room temperature was designed.

*3.4. Contrast Test Under Room Temperature*

New five identical Pb-free circuit boards were put into shock test with the same profile, but under normal temperature (25 °C) as contrasted. The test condition was 100G acceleration, 0.5-millisecond duration, 125 cm/s velocity change, and half-sine pulse. The circuit board was fixed on the test bench and subjected to a total of 12 shocks, which were two shock pulses of the peak acceleration, velocity change, and pulse duration in each of the positive and negative directions of three orthogonal axes (X, Y, and Z). All five circuit boards worked normally after the contrast test. There was no deformation and crack on solder balls, as shown in Figures 6 and 7. In the meanwhile, the tensile test and SEM analysis were also conducted, and the result is shown in Figure 8. Typical dimple fracture surface could be recognized, and it was an obvious ductile fracture surface that was different from Figure 5. The profile of the contrast test only changed the temperature compared with the initial cryogenic and shock test. Thus, we believed that cryogenic temperature made an impact on solder joints.

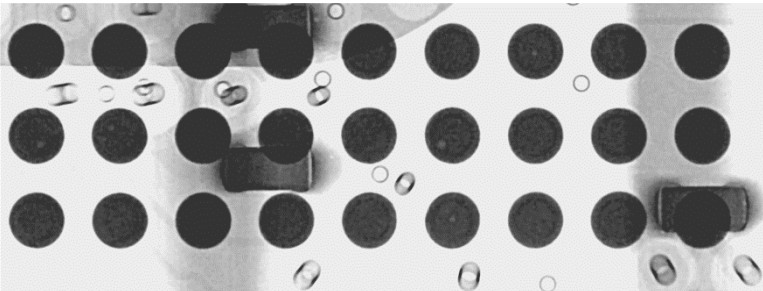

**Figure 6.** The X-ray detection result of an intact sample.

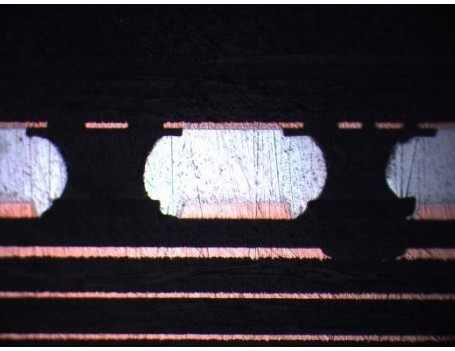

**Figure 7.** Intact solder joint observed under a metallurgical microscope.

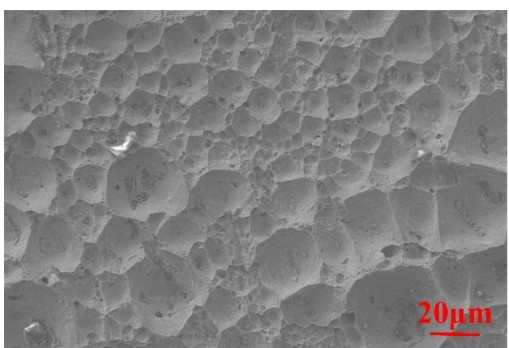

**Figure 8.** The contrast surface morphology of BGA under SEM at room temperature shows the ductile fracture.

*3.5. Failure Mechanism of BGA Solder Joints*

According to the previous analysis, the mechanical properties of solder joints at low-temperature are quite different from those at room temperature. It is the characteristic of Sn-rich material. The change of properties at ultra-low temperature will greatly influence the reliability of electronic components.

Through the series of tests and analysis, the failure mechanism of the Pb-free PCB could be confirmed. At low temperature ($-100$ °C), the fracture behavior of SAC305 changed from ductile to brittle. Meanwhile, because of the stress caused by shock test, cracks occurred in solder balls, and finally, the failure occurred. Aiming at SAC305, there was still a remaining problem. At what temperature would SAC305 material property change? To find the transition temperature to guide actual operation, further tests need to be carried out.

## 4. Failure Mechanism Verification

In order to verify the failure mechanism and find the transition temperature of SAC305, a low-temperature tensile test was designed, and corresponding samples were made and experienced the test process. Then, SEM analysis was used to confirm the material characteristic.

*4.1. Experiments*

Through a large number of experiments, researchers have found that when the temperature gradually decreases, the ductile fracture strength of different solder material significantly increases [29]. When the temperature reaches the transition temperature range, the fracture characteristic of solders changes obviously, as well as the energy required for fracture. The ductile fracture would transform into brittle fracture [30].

According to the previous researches, and because we could judge the ductileness/brittleness through tensile test data and fracture surface morphology, a low-temperature tensile test was conducted. The experimental scheme for obtaining the cryogenic mechanical property parameters of SAC305

solder joints were designed and indicated in Table 1. Tensile tests at different temperatures were carried out by the universal material testing machine.

At every specific temperature, the immersion time of samples was 0.5 h. Afterward, the tensile test was started with 0.01/s tension rate. The values of displacement from maximum tensile stress to complete fracture and tensile strength could be recorded as representative of the mechanical property of SAC305 material.

**Table 1.** Test temperature and corresponding samples.

| Number of Samples | Temperature (°C) |
|---|---|
| 1#, 2# | 25 |
| 3#, 4# | −50 |
| 5#, 6# | −70 |
| 7#, 8# | −80 |
| 9#, 10# | −100 |

### 4.2. Sample Introduction

According to the design of the low-temperature tensile test, corresponding samples of SAC305 material and clamp for the tensile test were designed. Figure 9 is the design drawing of the sample [31]. According to the experimental requirements for obtaining the low-temperature mechanical characteristics of the solder joint, Cu was selected as the base material for welding. SAC305 was used to connect the two Cu pieces. According to Figure 9, the length of the welding was 6 mm, the width was 1 mm, and the thickness of the test sample was 1 mm. This kind of design provided convenience for subsequent tests. Before welding, solder resist was applied to the surface of Cu blocks except for the weld area. Then, the two blocks were put into the clamp, and the solder paste was applied between the weld interface of two blocks. In order to simulate the actual welding process, a reflow welding temperature profile was set, and the samples were welded through the heating platform. Holding and welding temperatures were 175 °C and 250 °C, respectively. The duration of each stage could be seen in Figure 10.

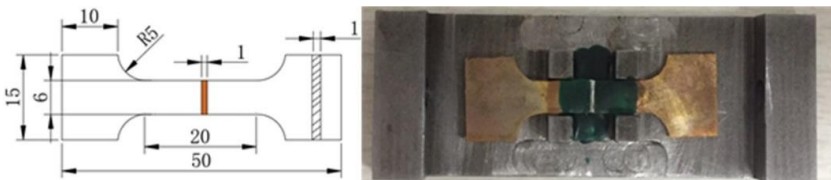

**Figure 9.** The design and the actual sample prepared for further tests.

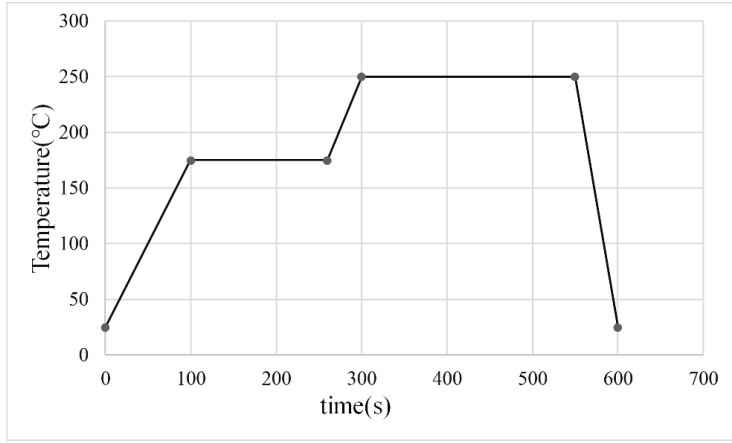

**Figure 10.** The welding temperature profile of sample preparation.

### 4.3. Results and Discussion

Through the test, the property parameters of solder joints at low temperature, the fracture surface samples, and the temperature range of ductile-brittle transition of solder joints could be obtained. Fracture strength and the displacement from the maximum tensile stress to the final fracture of Pb-free solder joints at different temperatures were obtained. Figure 11 shows the tensile strength and displacement curves of SAC305. According to the graphs and focus on the curve after the maximum tensile strength, it could be seen that the fracture mode of the solder joints was a ductile fracture at 25 °C. The slope of the curve after fracture changed slowly. With the decrease of temperature, slopes of the curve after the maximum tensile strength increased suddenly, meaning the mechanism changed from ductile fracture to brittle fracture. The transition temperature range of SAC305 solder joints was −70−−80 °C.

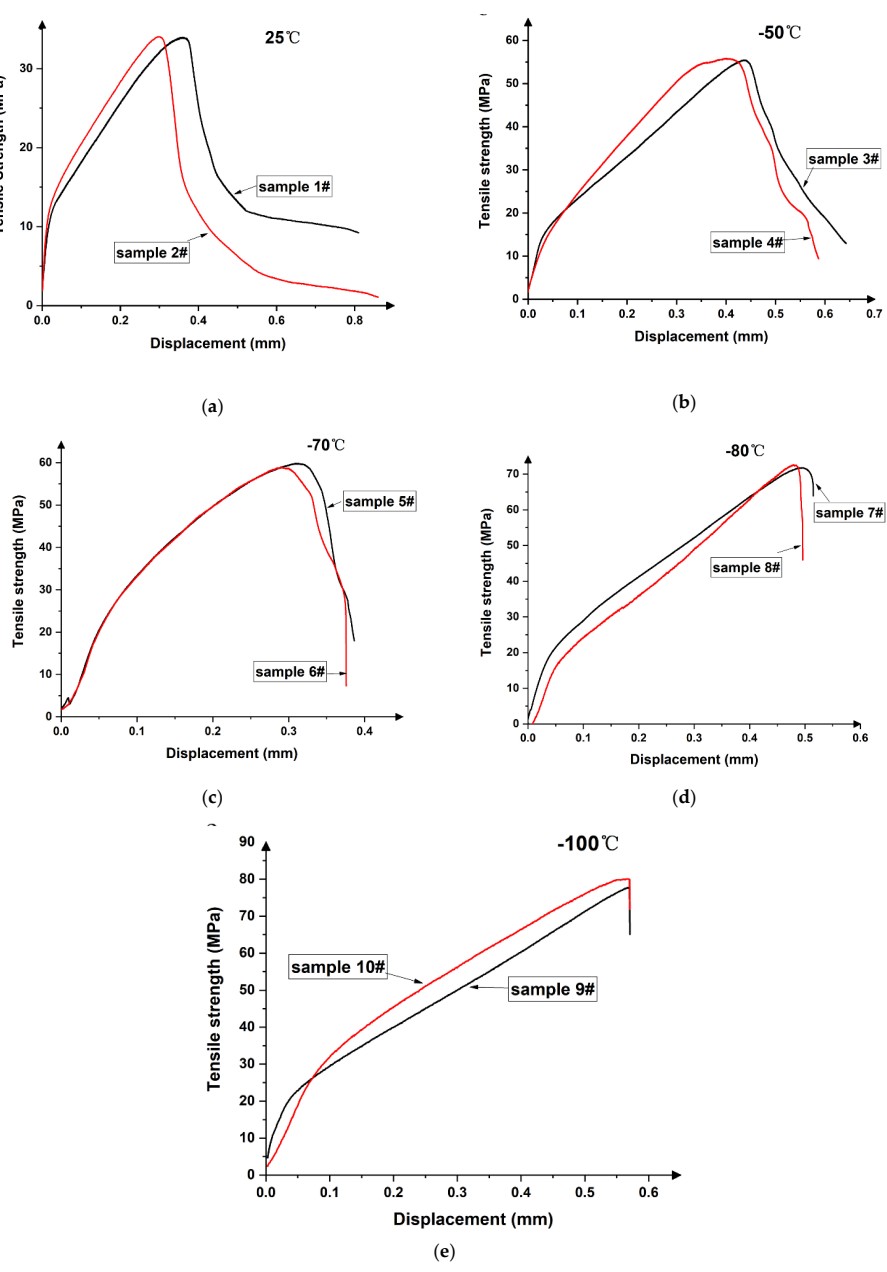

**Figure 11.** Tensile strength-displacement curve of Sn3.0Ag0.5Cu solder joints at different temperatures (**a**) 25 °C; (**b**) −50 °C; (**c**) −70 °C; (**d**) −80 °C; (**e**) −100 °C.

Table 2 shows the data of displacement from maximum tensile stress to complete fracture. This parameter is the characterization of the material toughness. The larger the value, the better the toughness. With the decrease in temperature, the values of displacement decreased as well. That is to say, the characteristic of ductile fracture became weaker, and the toughness of SAC305 was worse. The relationship between the toughness and the displacement from maximum tensile stress to complete fracture could be explained as: when the toughness decreased, the displacement decreased as well. The shock resistance of SAC305 gradually decreased. With the brittle fracture stage, when the external stress reached the fracture limit, small displacement could cause the fracture of the solder joint.

**Table 2.** Displacement from maximum tensile stress to complete fracture of Sn3.0Ag0.5Cu solder joint at different temperatures.

| Temperature (°C) | Displacement from Maximum Tensile Stress to Complete Fracture (mm) | | |
|---|---|---|---|
| | Sample 1 | Sample 2 | Average |
| 25 | 0.577 | 0.601 | 0.589 |
| −50 | 0.134 | 0.152 | 0.143 |
| −70 | 0.082 | 0.076 | 0.079 |
| −80 | 0.033 | 0.023 | 0.028 |
| −100 | 0.009 | 0.008 | 0.009 |

Figures 12–16 are the SEM graphs of the fracture surface at different temperatures. At 25 °C, −50 °C, and −70 °C, dimples were shown in the fracture surface morphology. It was obvious that the fracture mechanism was a ductile fracture. At −80 °C and −100 °C, the figures showed the mechanical changes to brittle fracture. Different surface morphology under lower temperatures showed different material fracture characteristics from that under higher temperatures. From Figures 15 and 16, intergranular fracture surface and river pattern could be perceived. They are the characteristics of brittle fracture. The results could be a supplementary instruction of the transition of material characteristics and its temperature range.

Through the analysis of test results, we could note that at cryogenic temperature, the fraction mechanism of SAC305 material changed from ductility to brittleness. The transition temperature range was −70–−80 °C. Though the strength of extension was greater at a lower temperature, the material brittleness was higher as well. That means if there are cracks in SAC305 solder joints, which have been verified to easily occurring after the Pb-free welding process [32], less stress can cause crack growth and even fracture of Pb-free solder joint at low temperature.

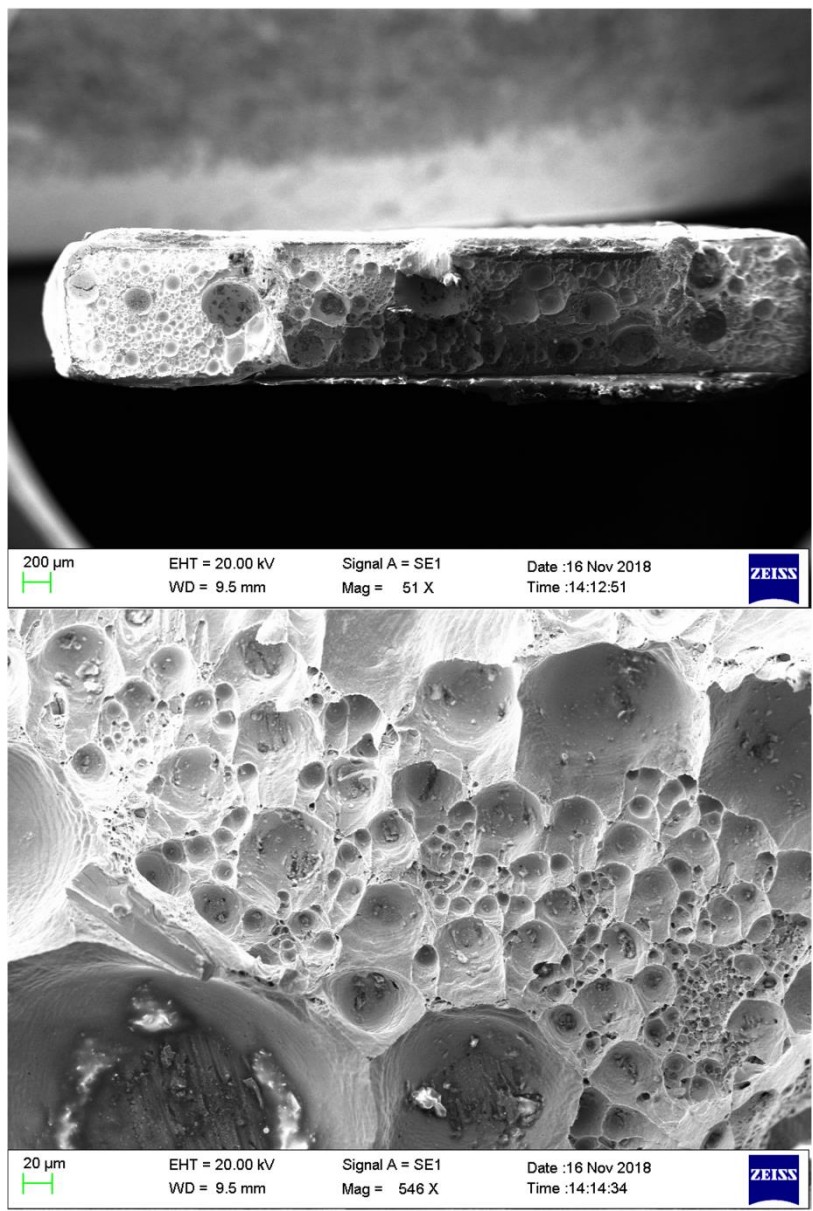

**Figure 12.** SEM graph of fracture surface at 25 °C (ductile).

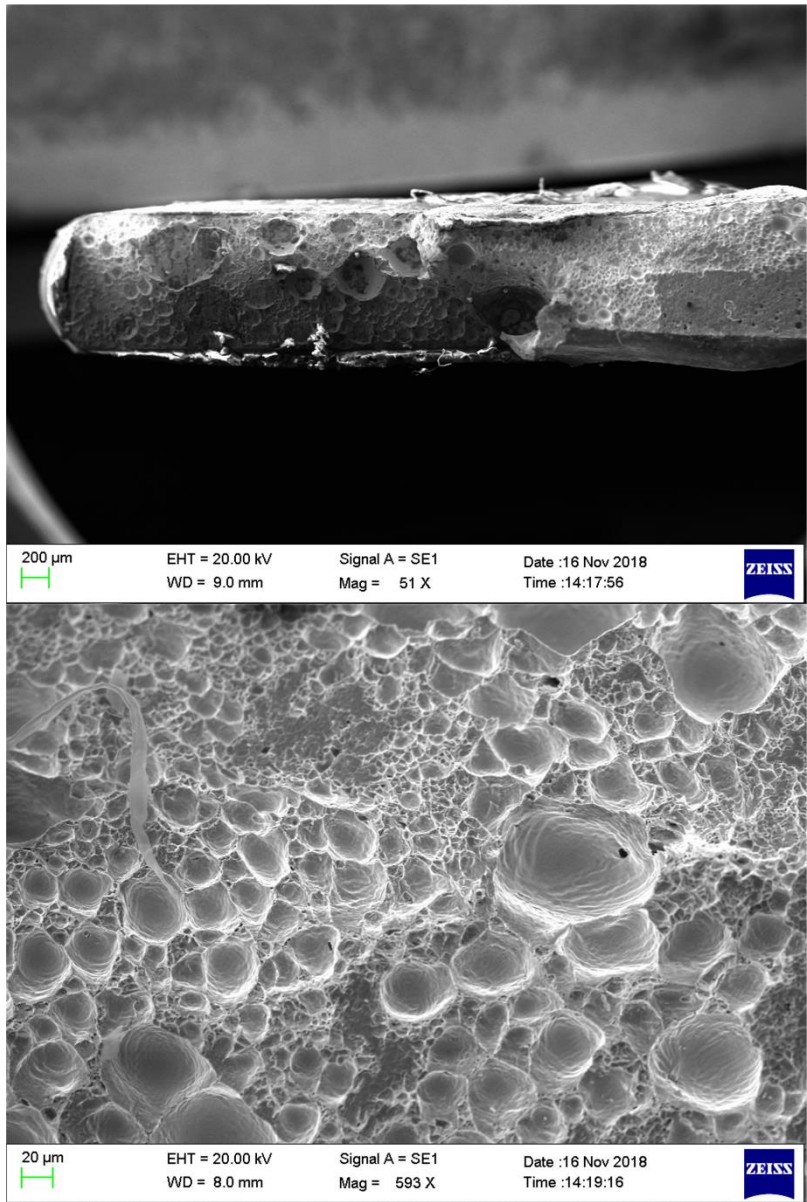

**Figure 13.** SEM graph of fracture surface at −50 °C (ductile).

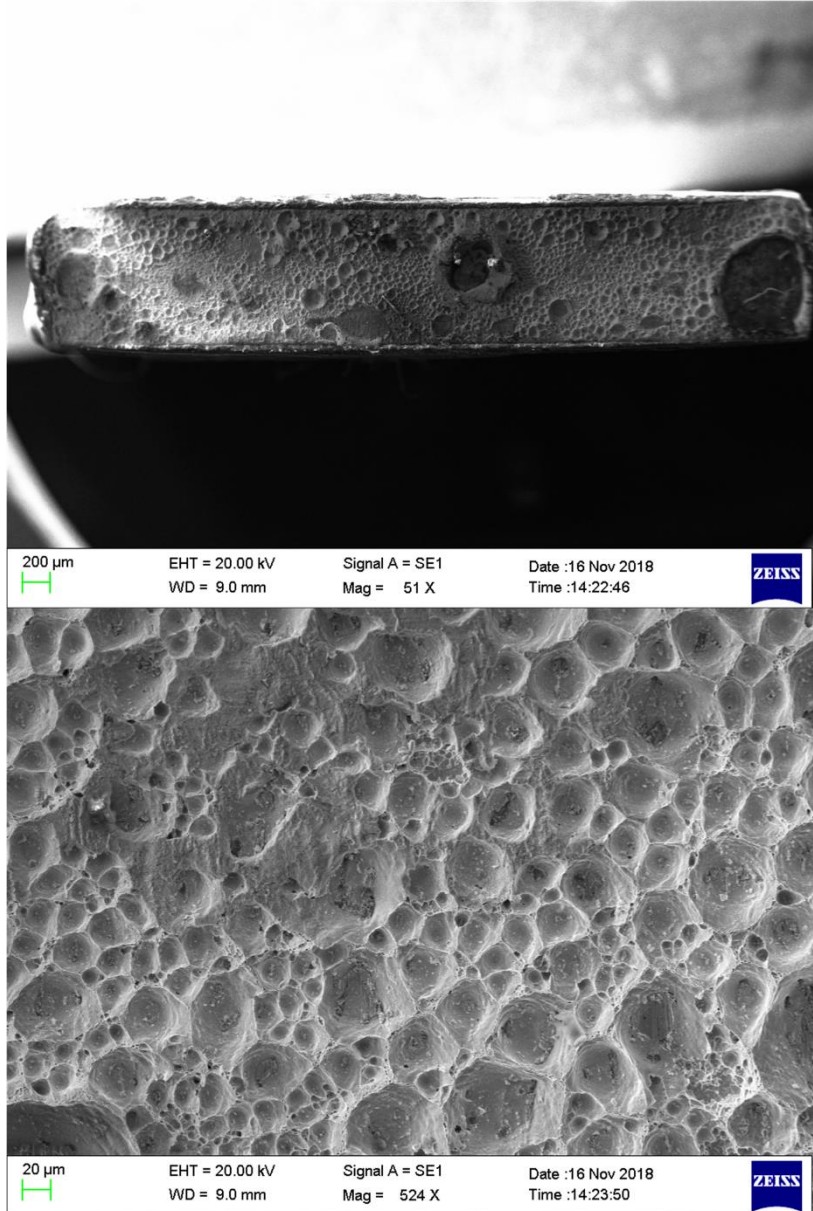

**Figure 14.** SEM graph of fracture surface at −70 °C (ductile).

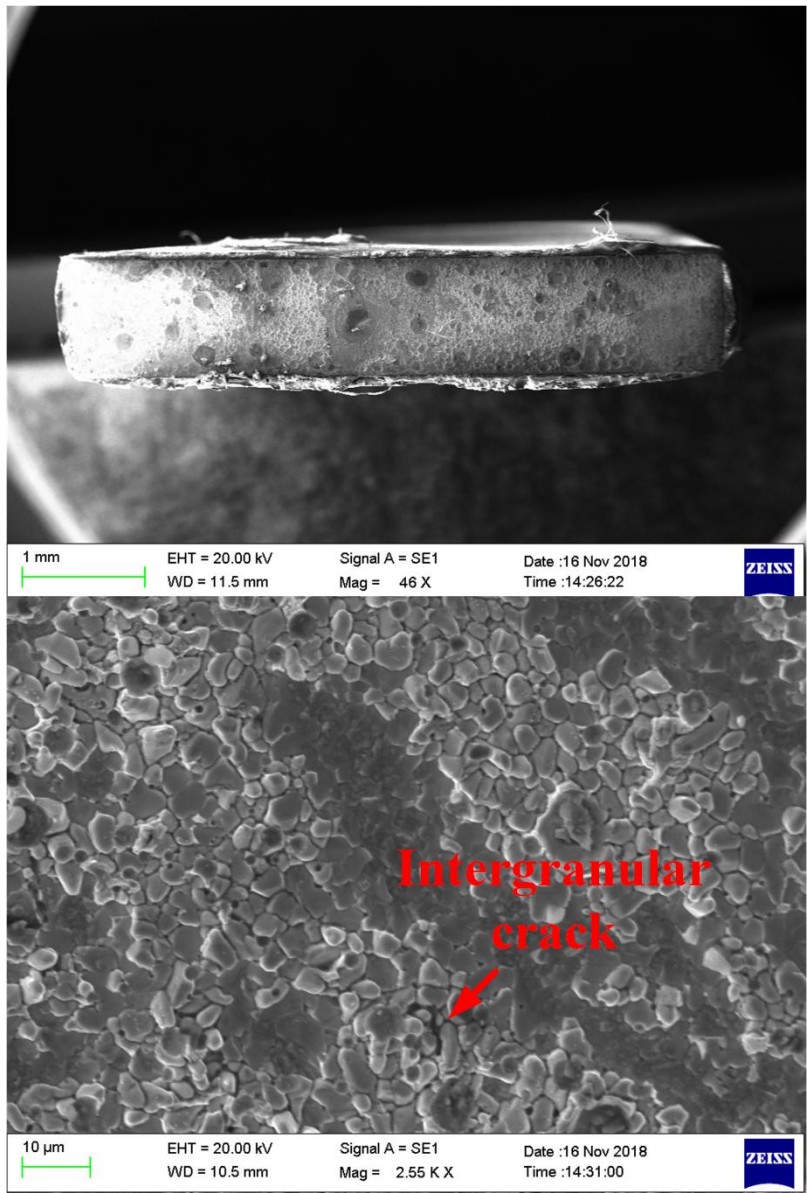

**Figure 15.** SEM graph of fracture surface at −80 °C (brittle).

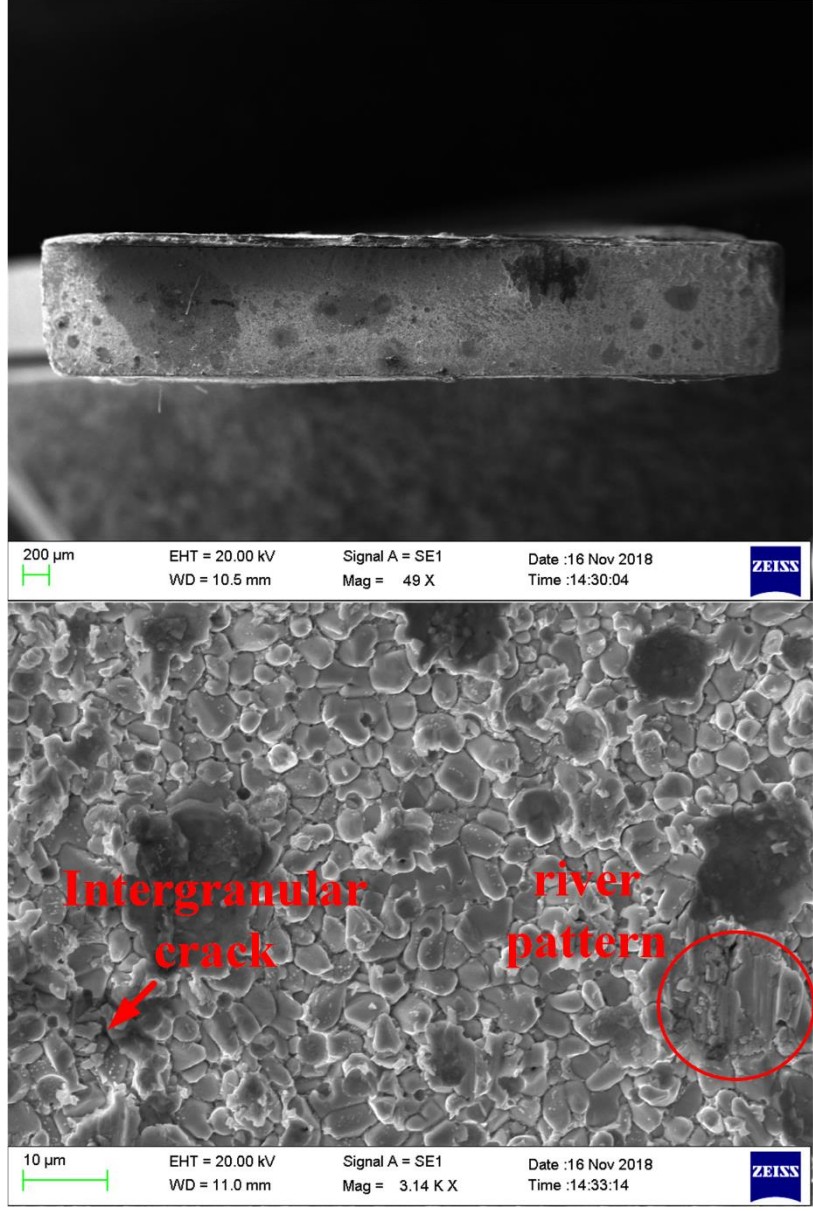

**Figure 16.** SEM graph of fracture surface at −100 °C (brittle).

## 5. Conclusions

In order to study the failure of a typical Pb-free circuit board, a series of methods was conducted to make failure analysis. Nondestructive examinations were used, and the failure position was located at the BGA solder joint. Then, destructive examinations were conducted, and the cause of failure was confirmed as the change of material characteristic under extremely low-temperature. A contrast test supported this result. Finally, verification experiments were conducted to verify the failure mechanism and find the transition temperature range of SAC305, which could make a guide for Pb-free circuits' deep space operation.

The results of failure analysis indicated the cause of failure. The Pb-free circuit board went through the shock tests at −100 °C. At this temperature, the fracture characteristic of SAC305 was brittleness, which was different from ductileness under room temperature. When the circuit underwent stress caused by shock test, it was easy for the solder joint to have cracks. When the crack grew to a specific length, the failure occurred. In the meanwhile, the transition temperature range of material property was also confirmed as −70—−80 °C.

Through this failure analysis case, it alerts us to focus attention on the reliability of Pb-free material applications for deep space exploration. Because of the low-temperature characteristics of SAC305, there may be a higher failure risk of actual operation. According to our study, the environment temperature needs to be kept higher than −70 °C. More studies need to be conducted to improve the reliability of Pb-free soldering used for aerospace components. Meanwhile, associated preventive methods, such as thermal preservation, are necessary.

**Author Contributions:** Conceptualization, B.W.; Methodology, Y.L.; Validation, Y.L.; Formal Analysis, G.F.; Investigation, Y.L.; Writing—Original Draft Preparation, Y.L.; Writing—Review and Editing, M.J.; Visualization, Y.L.; Supervision, W.Z. and X.Y. All authors have read and agreed to the published version of the manuscript.

**Funding:** This research received no external funding.

**Acknowledgments:** The authors are thankful to School of Reliability and Systems Engineering, Beihang University for providing the test equipment.

**Conflicts of Interest:** The authors declare no conflict of interest.

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
