# Peer review of "Failure Analysis of SAC305 Ball Grid Array Solder Joint at Extremely Cryogenic Temperature"

_applsci, doi:10.3390/app10061951_

Round 1

Reviewer 1 Report

The paper deals with failure analysis of SAC305 ball grid array solder joint at extremely cryogenic temperature. This research is focused for the aerospace and space exploration idustry. The study of reliability of solder joints in electronic parts for this kind of industry is very current nowadays.

The authors showing very interesting and important results. 

However, there are some comments that needs to be considered:

Introduction section:

  • There are exceptions for the use of lead in case of technological advances in development. These exceptions include space exploration according the European Union directive known as RoHS (Restriction of Hazardous Substances). Although the idea of total elimination of lead from solder alloys is very wise, it would be good to mention these exceptions in the introduction section.
  • Authors mentioned Sn-Ag-Cu solders as most useful materials as a replacement of Sn-Pb solder. However, only SAC305 is mentioned. The authors should mention also other solder composition in group of SAC solders. The development of SAC type solders is also oriented towards adding a small amount of other alloying elements to improve their properties. Such as in this study:
    KOLEŇÁK, Roman - AUGUSTIN, Robert - MARTINKOVIČ, Maroš - CHACHULA, Michal. Comparison study of SAC405 and SAC405+0.1% Al lead free solders. In Soldering and Surface Mount Technology. Vol. 25, Iss. 3 (2013), pp. 175-183

Subject of study and failure background:

  • Did you think about the space environment such as cosmic radiation which also affect the solder joint reliability? 

Failure analysis of the memory:

  • Please increase the size of the figure 3. The deformation of the solder ball is not clearly visible.

Failure mechanism verification:

  • Please correct the image numbering from section 4.3.

The paper is well organized and gives comprehensive knowledge of SAC 305 solder for such applications. The results idicated that this solder may not be suitable for the space exploration misions because of its higher failure risk. Based on this work, the future research should be focused to the development of other solder alloys for space exploration applications.

After minor revision, I recommend the paper for publication in this journal.

Reviewer 2 Report

In this paper, the authors verified the reliability of SAC305 Pb-free solder joint for use in an environment of cryogenic temperature such as an aerospace field. The authors analyzed the failure sample which was actually used for aerospace application and verified the failure mechanism by conducting some experiments. I thought that this paper is suitable for being considered as a publication in Applied Sciences. However, the improvement of this paper is needed. Therefore, the authors have to make a revision and supplement before considering accept this paper. Accordingly, I recommend that this paper should be major revisions. I hope following comments would be helpful for improving the manuscript.

In addition to the reasons above, there are some questions and comments as below.

#1. English needs some improvements.

#2. In page 2, line 51-52: The SAC305 was first mentioned. The composition of SAC305 are needed to firstly mentioned such as Sn–3.0Ag–0.5Cu.

#3. In page 3-4 and line 102 (page 3)-109(page 4): The authors mentioned that the deformation of solder balls were obviously seen in an X-ray image of Fig. 3. However, this reviewer thought that Fig. 3 image was not clear. Therefore, as you can, the authors revise the image by using the magnified images to easily distinguish deformed solder balls and no-deformed solder balls.

#4. In page 4, line 118-123: Please add the condition of tensile test using a universal tensile testing machine.

#5. In page 4, Fig. 5: The scale bar in Fig. 5 was not clear.

#6. In page 4, line 129-131: The authors mentioned the word of ‘isomer transition’. Does the ‘isomer transition’ mean the allotropic transformation such as tin-pest?

#7. In page 5, line 135: Does the shock test mentioned mean the cryogenic test?

#8. In page 5, line 138: Please clearly explain the procedures of shock test

#9. In page 5, line 144: What does qualification test mean?

#10. In section 3.4.: It is need to add clear explanation.

#11. In Fig. 6: It is need to clearly distinguish Fig. 3. and Fig. 6.

#12. In Fig. 8: Please make the scale bar clearly visible.

#13. In Fig. 6, line 159: What does the shock force mean?

#14. In page 6, line 180: Please re-check the unit of tension rate. Is a rate of 0.01/s correct?

#15. In section 4.2: What is the joining process to weld the two blocks. Please add explanation of process such as a method or joining temperature.

#16. In section 4.3, page 7: The authors showed the results of the displacement and strength after the tensile test. Did you compared toughness? As you know, the toughness is also important. As you can, I think that the effect on the toughness is also worth considering for this paper.

17. In page8-9: Some figure number was wrong. Please check. I also thought that scale bars in figures were not clear. Please revise.

Round 2

Reviewer 2 Report

This reviewer found that authors made the revisions and received that authors’ response letter in accordance with the comments of this reviewer. However, some improvements are needed for the publication. Accordingly, I recommend this paper to be published of the Applied Sciences after minor revision.

In addition to the reasons above, there are some comments as below.

#1. Some expression are still needed to revise. For example, the authors used a mixture of element symbols (Pb etc.) and element names (lead etc.). In page 1 and line 43, English capital letter error was use like Tin. Besides, some errors were observed through this manuscript. Please re-check English.

#2. In abstract, like previous comment, SAC305 expression was firstly mentioned.

#3. In Fig. 12~15, Are the scale bar in these figures right? Please re-check.
